# L-DOPA modulates activity in the vmPFC, nucleus accumbens, and VTA during threat extinction learning in humans

Roland Esser[1], Christoph W Korn[1,2], Florian Ganzer[1,3], Jan Haaker[1]*

[1]Department of Systems Neuroscience, University Medical Center Hamburg-Eppendorf, Hamburg, Germany; [2]Section Social Neuroscience, Department of General Psychiatry, Heidelberg, Germany; [3]German Center for Addiction Research in Childhood and Adolescence, University Medical Center Hamburg-Eppendorf, Hamburg, Germany

**Abstract** Learning to be safe is central for adaptive behaviour when threats are no longer present. Detecting the absence of an expected threat is key for threat extinction learning and an essential process for the behavioural treatment of anxiety-related disorders. One possible mechanism underlying extinction learning is a dopaminergic mismatch signal that encodes the absence of an expected threat. Here we show that such a dopamine-related pathway underlies extinction learning in humans. Dopaminergic enhancement via administration of L-DOPA (vs. Placebo) was associated with reduced retention of differential psychophysiological threat responses at later test, which was mediated by activity in the ventromedial prefrontal cortex that was specific to extinction learning. L-DOPA administration enhanced signals at the time-point of an expected, but omitted threat in extinction learning within the nucleus accumbens, which were functionally coupled with the ventral tegmental area and the amygdala. Computational modelling of threat expectancies further revealed prediction error encoding in nucleus accumbens that was reduced when L-DOPA was administered. Our results thereby provide evidence that extinction learning is influenced by L-DOPA and provide a mechanistic perspective to augment extinction learning by dopaminergic enhancement in humans.

*For correspondence:
j.haaker@uke.de

Competing interest: The authors declare that no competing interests exist.

## Introduction

In order to thrive in dangerous environments, it is important to know when threats are disappearing and situations become safe. As such, safety learning is central for adaptive behaviour and deficits characterise symptoms in a wide range of anxiety-related disorders (*Duits et al., 2015*; *Fenster et al., 2018*; *Lissek et al., 2005*; *Milad and Quirk, 2012*). Yet, the neuropharmacological mechanism to augment safety learning by encoding the absence of potential threats or aversive outcomes in humans are not completely understood.

Safety learning is often investigated in laboratory protocols of extinction training. Here, a learned predictor (conditioned stimulus, CS) for an aversive outcome (unconditioned stimulus, US) is turning into a safety signal when the expected aversive outcome is omitted. This omission of the expected US after CS presentation is thought to drive extinction or safety learning. Extinction learning is held to involve learning of a new CS–no US association that inhibits the acquired CS–US association (*Bouton, 2004*; *Bouton and Nelson, 1994*). However, it is only incompletely understood which neural system in humans detects the omission of the expected aversive outcome and, hence, initiates a shift from threat to safety. Studies in *Drosophila* (*Felsenberg et al., 2018*) and rodents (*Badrinarayan et al., 2012*; *Correia et al., 2016*; *Holtzman-Assif et al., 2010*; *Luo et al., 2018*; *Mueller et al., 2010*;

*Salinas-Hernández et al., 2018*; *Zhang et al., 2020*; *Aksoy-Aksel et al., 2021*) revealed that the omission of an expected aversive outcomes depends on signals in the dopaminergic system. In rodents, this involved dopaminergic neurons in the ventral tegmental area (VTA), the nucleus accumbens, and the medial prefrontal cortex, as well as projections between the VTA and nucleus accumbens (*Badrinarayan et al., 2012*; *Luo et al., 2018*; *Oleson et al., 2012*; *Rodriguez-Romaguera et al., 2012*). Importantly, these neural regions were also found to underpin the processing of rewarding outcomes. When signalling rewards, this system does not simply detect a rewarding outcome, but codes a difference between the expected reward and the actual outcome in form of an expectancy violation or prediction error (*Schultz et al., 1997*). In other words, reward-related response in the VTA, nucleus accumbens, and vmPFC reflect outcomes that are better than expected.

Similarly, the omission of an expected aversive US, which could be framed as 'better than expected', might well be under the influence of a dopaminergic signal: At the time-point of US omission, a dopaminergic system might encode an expectancy violation that signals the difference between the expected aversive US and the omitted aversive outcome (for review, see *Abraham et al., 2014*; *Kalisch et al., 2019*; *Nasser and McNally, 2012*; *Sartori and Singewald, 2019*).

Even though this idea has not been formally tested, it is supported by two functional neuroimaging study in humans. These studies provided initial evidence that computational modelling of an prediction error for the omitted aversive outcome during extinction training involves activity in the nucleus accumbens (*Thiele et al., 2021*; *Raczka et al., 2011*) and that this activity was modulated by a genetic variance of the dopamine transporter gene (*Raczka et al., 2011*). Additionally, there is cross-species evidence for enhanced extinction memory consolidation by augmented dopaminergic transmission after extinction training (by administration of L-DOPA [*Gerlicher et al., 2018*; *Haaker et al., 2015*; *Haaker et al., 2013*]). These latter studies suggest that dopaminergic enhancement of extinction memory retrieval is mediated by augmenting activity in the ventral part of the medial prefrontal cortex (vmPFC), a structure that is central for extinction learning and memory retrieval (*Kalisch et al., 2006*; *Milad et al., 2007*; *Milad and Quirk, 2002*; *Phelps et al., 2004*). It is, however, not clear if enhancing of dopaminergic neurotransmission would strengthen extinction *learning* by modulating vmPFC activity.

In this study, we tested if extinction learning is associated with activity changes in the vmPFC and if such activity is modulated by administration of the dopaminergic precursor L-DOPA. We further tested if the unexpected omission of the US during extinction learning is coded in midbrain pathways that connect the VTA and the nucleus accumbens and if activity within this pathway is modulated by L-DOPA. Based on previous studies (*Gerlicher et al., 2018*; *Haaker et al., 2013*), we hypothesised that L-DOPA administration before extinction training would decrease threat responses at retention tests.

## Results

### Behavioural and physiological outcome measures

#### Acquisition of CS–US contingencies on day 1

Participants in both groups learned CS–US contingencies during acquisition training, which was indicated by a CS-type main effect that consisted of enhanced responses to the CS+ as compared to the CS− in all dependent measurements, namely binary (yes/no), trial-wise US expectancy ratings (CS-type main effect in rmANOVAs: US expectancy $F(1,44) = 203.9$, $p<0.001$, $\eta_p^2 = 0.823$, mean difference: $0.578 \pm 0.660/0.496$ [95% CI], see *Figure 1a*), SCR ($F(1,43) = 41.7$, $p<0.001$, $\eta_p^2 = 0.493$, mean difference: $0.088 \pm 0.115/0.061$ [95% CI], see *Figure 1b*) and fear ratings: ($F(1,44) = 116.0$, $p<0.001$, $\eta_p^2 = 0.725$, mean difference: $0.361 \pm 0.428/0.294$ [95% CI]; see *Figure 1c*), see *Supplementary file 1b* for full statistics, means and CI. Unexpectedly, we found an interaction effect in US expectancy between CS-type, trial, and group-status (i.e., subjects that were allocated to receive Placebo or L-DOPA on the next day: CS-type × trial × group $F(2,88) = 3.3$, $p=0.044$, $\eta_p^2 = 0.07$). However, follow-up group comparisons of block-wise US expectancy did not support any differences in CS+ or CS− responses (two-tailed independent post-hoc t-tests: p-values [FWE ] >0.255, see *Supplementary file 1c*) or CS+/CS− discrimination between groups (p(FWE) > 0.65, CS discrimination was descriptively lower in the prospective Placebo vs. L-DOPA group, see *Supplementary file 1c*). There was no support

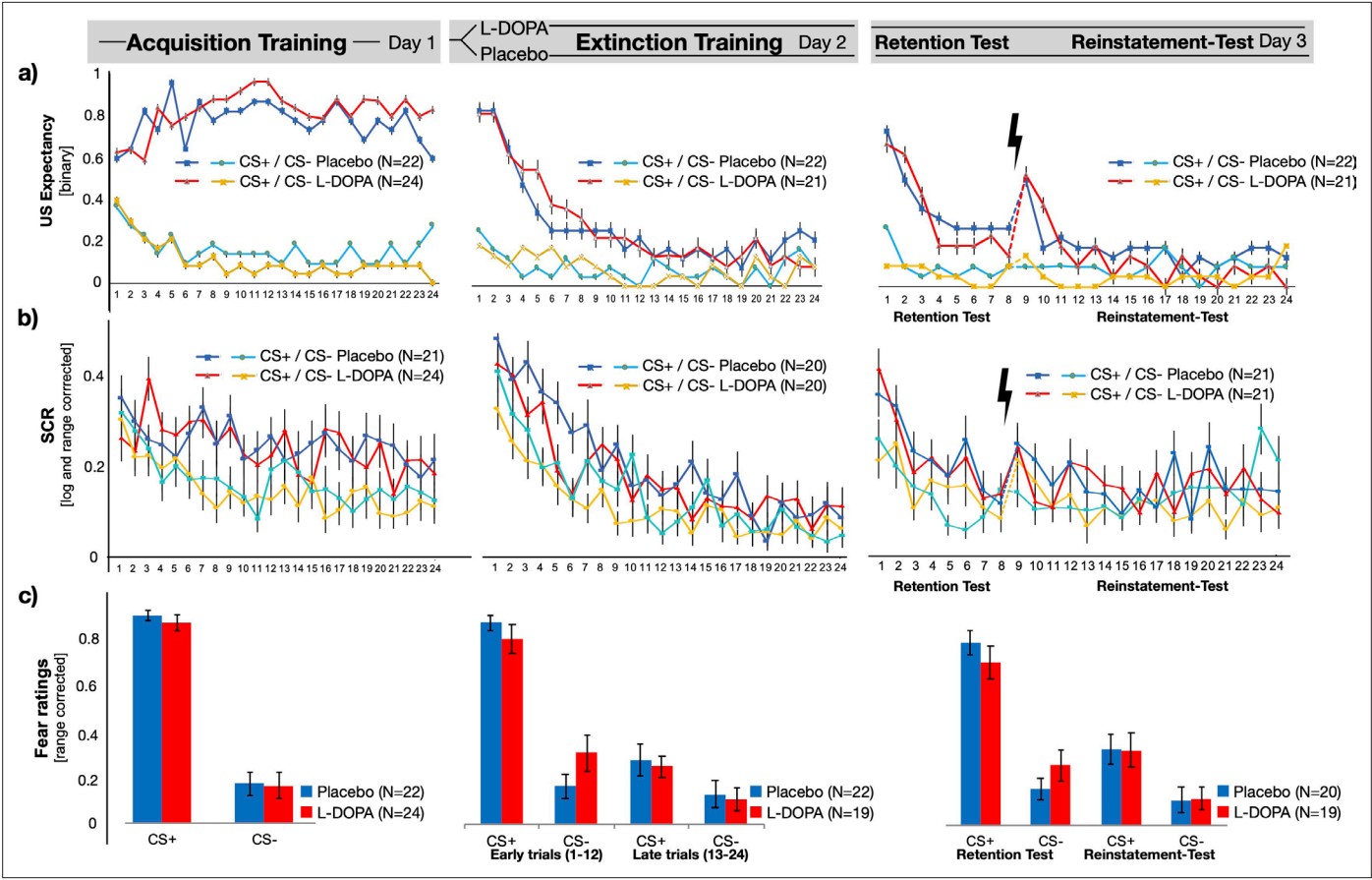

**Figure 1.** Behavioural and psychophysiological outcome measures on days 1, 2, and 3. (**a**) US expectancy, (**b**) SCR, and (**c**) fear ratings reflect successful acquisition of CS–US contingencies during acquisition and decreasing responses during extinction training. Retention of CS-US memory was evident during retention test on day 3, as well as initial enhancement of responses after reinstatement within three trials after presentation of the reinstatement USs. Differential SCRs (CS+ − CS−) in three trials after reinstatement were lower in the L-DOPA, when compared to the Placebo group (see *Figure 2*). SCR = skin conductance responses, CS = conditioned stimulus.

for differences between groups in fear ratings or SCRs (group main effect or interaction ps>0.1, see *Supplementary file 1b*).

## Extinction learning on day 2

On day 2, participants discriminated between CS+ and CS−, as indicated by a main effect of CS-type across all outcome measures (CS-type main effect in rmANOVAs: US expectancy: F(1,41)=22.3, p<0.001, $\eta_p^2$ = 0.353, mean difference: 0.183 ± 0.269/0.106 [95% CI], SCR: F(1,38)=23.9, p<0.001, $\eta_p^2$ = 0.386, mean difference: 0.065 ± 0.092/0.039 [95% CI], and fear ratings: F(1,41)=61.83, p<0.001, $\eta_p^2$ = 0.601, mean difference: 0.345 ± 0.434/0.256 [95% CI]; see Table *Supplementary file 1d* for full statistics). Responses in all measures decreased over the time course of extinction training (CS-type by block interaction, all ps<0.05, see *Supplementary file 1d*, see *Figure 1*). In particular, trial-wise US expectancy ratings indicated successful extinction learning of the CS–US association, i.e. differential CS responses in the first two blocks (CS+ > CS−, Block 1: p<0.001, Block 2 p=0.048), but not the last (Block 3: p=0.57, see *Supplementary file 1d*).

Importantly, the analyses of fear rating indicated only a weak support for an interaction between CS-type, block, and group (F(1,42)=3.884, p=0.059, $\eta_p^2$ = 0.095). In accordance with our hypothesis, we found lower differential ratings of fear (CS+ − CS−) at the beginning of day two in the L-DOPA group when compared to placebo controls, but this difference was not supported when correcting for multiple comparisons (one-sided, post hoc independent t-test: L-DOPA < Placebo, t(41)=1.911, p(uncorr)=0.032, p(FWE-corr)=0.064, Cohen's d: –0.583, L-DOPA mean: 0.430 ± 0.523 [SD], Placebo mean: 0.682 ± 0.322 [SD], see *Supplementary file 1e*). Exploratory analyses suggested that this

effect might be driven by lower ratings to the CS+ and the extinction context (presented as the ITI) in the L-DOPA group, but none of these comparisons survived correction for multiple testing (p-values (FWE) > 0.256, see *Supplementary file 1e*, see supplementary results for fear ratings without range-correction). Hence, we found no statistical support for reduced differential fear appraisal of the CSs in the L-DOPA group, as compared to the Placebo group. We further found no statistical evidence that would support a difference between groups in US expectancy ratings or SCR (see *Supplementary file 1d*).

Next, we examined how decreasing US expectancy, which indicates extinction learning, is driven by expectancy violation from the omission of the US, by fitting US expectancy ratings with a Rescorla-Wagner–Pearce-Hall hybrid model (*Boll et al., 2013*; *Li et al., 2011*). The fitted prediction error (as a measure of expectancy violation), associability (as a measure of prediction error-guided surprise) and learning rate did not differ between groups (two-sided independent sample t-test for mean prediction error: t(40)=0.097, p(uncorr) = 0.923, p(FWE)>0.99; mean associability: t(40)=0.015, p=0.988, and mean learning rate: t(40)=0.179, p(uncorr)=0.859, p(FWE)>0.99; see *Supplementary file 1f*).

### Memory retrieval on day 3

Retrieval was tested on day three within an generalisation context that consisted of a mixture of the acquisition and extinction context, i.e. one context that entailed 50 % of the furniture from the acquisition context A and 50 % from the extinction context B (*Andreatta et al., 2015*), which also involves contextual renewal of conditioned threat responses (*Vervliet et al., 2013*). Participants discriminated between CSs in all outcome measures (CS-type main effect in rmANOVAs: US expectancy: F(1,41)=23.21, p<0.001, $\eta_p^2 = 0.361$, mean difference: 0.253 ± 0.358/0.148 [95% CI], SCR: F(1,40) = 24.07, p<0.001, $\eta_p^2 = 0.376$, mean difference: 0.076 ± 0.108/0.045 [95% CI], and fear ratings: F(1,41) = 54.79, p<0.001, $\eta_p^2 = 0.578$, mean difference: 0.512 ± 0.652/0.372 [95% CI]; see *Supplementary file 1g*). US expectancy ratings further indicated a general reinstatement of CS+ and CS− responses, when comparing the last three trials before and after the reinstatement USs (see *Supplementary file 1i*), but not within a block-wise reinstatement analyses (see *Supplementary file 1h*).

Importantly, the SCR analyses of the three trials before and after reinstatement revealed a difference between groups in differential CS responses (CS-type by group interaction F(1,40)=5.443, p=0.025, $\eta_p^2 = 0.120$, see *Figure 2* and *Supplementary file 1i*), indicating lower CS discrimination in the L-DOPA group when compared with the Placebo controls after the reinstatement procedure (one-sided, L-DOPA < Placebo post hoc t-test: t(40)=2.405, p(FWE-corrected)=0.020, Cohen's d = −0.741, L-DOPA mean: −0.006 ± 1.31 [SD], Placebo mean: 0.086 ± 0.116 [SD], see *Figure 2—figure supplement 1* and *Supplementary file 1j*). Post hoc comparisons of CS+ and CS− responses between groups did not support a difference between L-DOPA and Placebo (p>0.19, see *Supplementary file 1k*). Additional analyses that included two to five trials also revealed a difference between groups in differential SCRs (see *Supplementary file 1l,m* and *Figure 2—figure supplement 2*). While our analysis revealed a difference between groups in differential SCRs, there was no strong support for difference between groups in CS+ or CS− responses.

While our a priory hypothesis was an effect of L-DOPA on the psychophysiological measurements at retrieval test on day 3, our analyses suggest that L-DOPA administration during extinction training reduced differential threat responses after reinstatement.

## Administration of L-DOPA enhances vmPFC responses reflecting decreasing US expectancy during extinction learning

First, our analyses of neural responses focused on the effect of L-DOPA on extinction learning, where we expected an involvement of the vmPFC that is modulated by L-DOPA. To this end, we examined brain regions that increased their activity to a decrease of US expectation. In order to examine extinction learning by decreasing US expectancy, we contrasted responses during extinction training to CS+ trials when participants expected no US against CS+ trials in which participants expected an US (i.e., expectation of no US > expectation of a US). We found that decreasing US expectancy was accompanied by more pronounced signalling in the right vmPFC in the L-DOPA group as compared to the placebo group (see *Figure 3a*). Thus, administration of L-DOPA augmented vmPFC activity during extinction learning, i.e., when participants decreased their US expectancy.

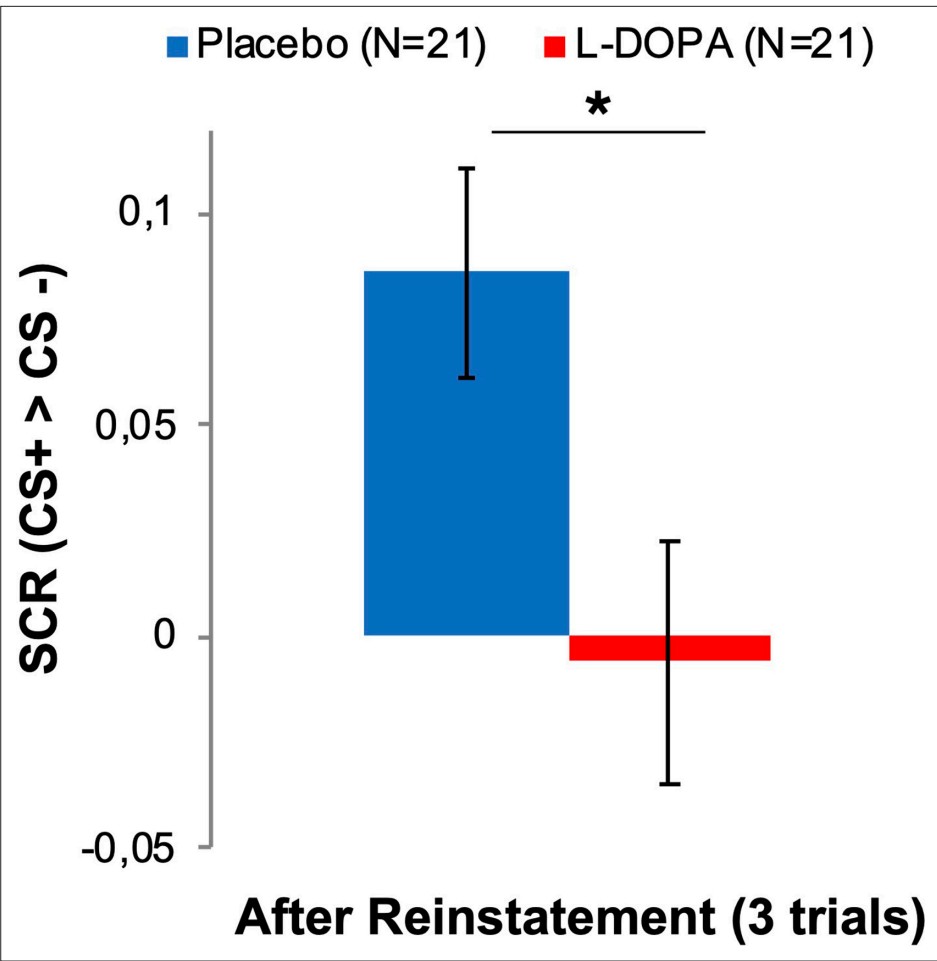

**Figure 2.** L-DOPA administration during extinction learning decreased differential SCRs after reinstatement on day 3. Differential SCRs (CS+ > CS−) were decreased when compared to the Placebo group within three trials after the reinstatement procedure (ANOVA: CS-type by group interaction). See *Figure 2—figure supplement 1l* for CS-specific and trial-wise responses. Additional analyses that include two to five trials revealed a difference between groups in differential SCRs (see *Supplementary file 1* and *Figure 2—figure supplement 2*). SCR = skin conductance responses, CS = conditioned stimulus.

The online version of this article includes the following figure supplement(s) for figure 2:

**Figure supplement 1.** L-DOPA administration during extinction learning decreases SCRs after reinstatement.

**Figure supplement 2.** Effect sizes for comparisons of differential CS responses between groups (one-sided unpaired t-test) before and after reinstatement.

Next, we tested if this difference in the right vmPFC activity was related to individual differences in the retrieval of conditioned threat responses. A previous study indicated that vmPFC activation during extinction learning was associated with retention of extinction memory (measured as differential SCR) 24 hr later (*Gerlicher et al., 2018*). Indeed, we found that higher vmPFC activation is associated with reduced differential SCR, which could indicate better individual extinction memory retention, 24 hr later (two-sided Pearson correlation: $t(38) = -2.18$, p-value=0.035, $r = -0.3273302$ [95% CI: −0.58 to 0.018, see *Figure 3b*]). Hence, vmPFC responses during extinction learning were elevated after L-DOPA administration, and such enhanced vmPFC activity is associated with reduced retrieval of differential threat responses (measured as SCR) 24 hr later. Importantly, there was no difference between groups detectable in SCRs during retrieval test, which might have biased this correlation (*Makin and Orban de Xivry, 2019*). However, it might be possible that L-DOPA treatment has an indirect effect on SCR during retrieval test, which was mediated by vmPFC activity during extinction learning. Indeed, we found support for a treatment effect of L-DOPA on SCRs during retrieval test,

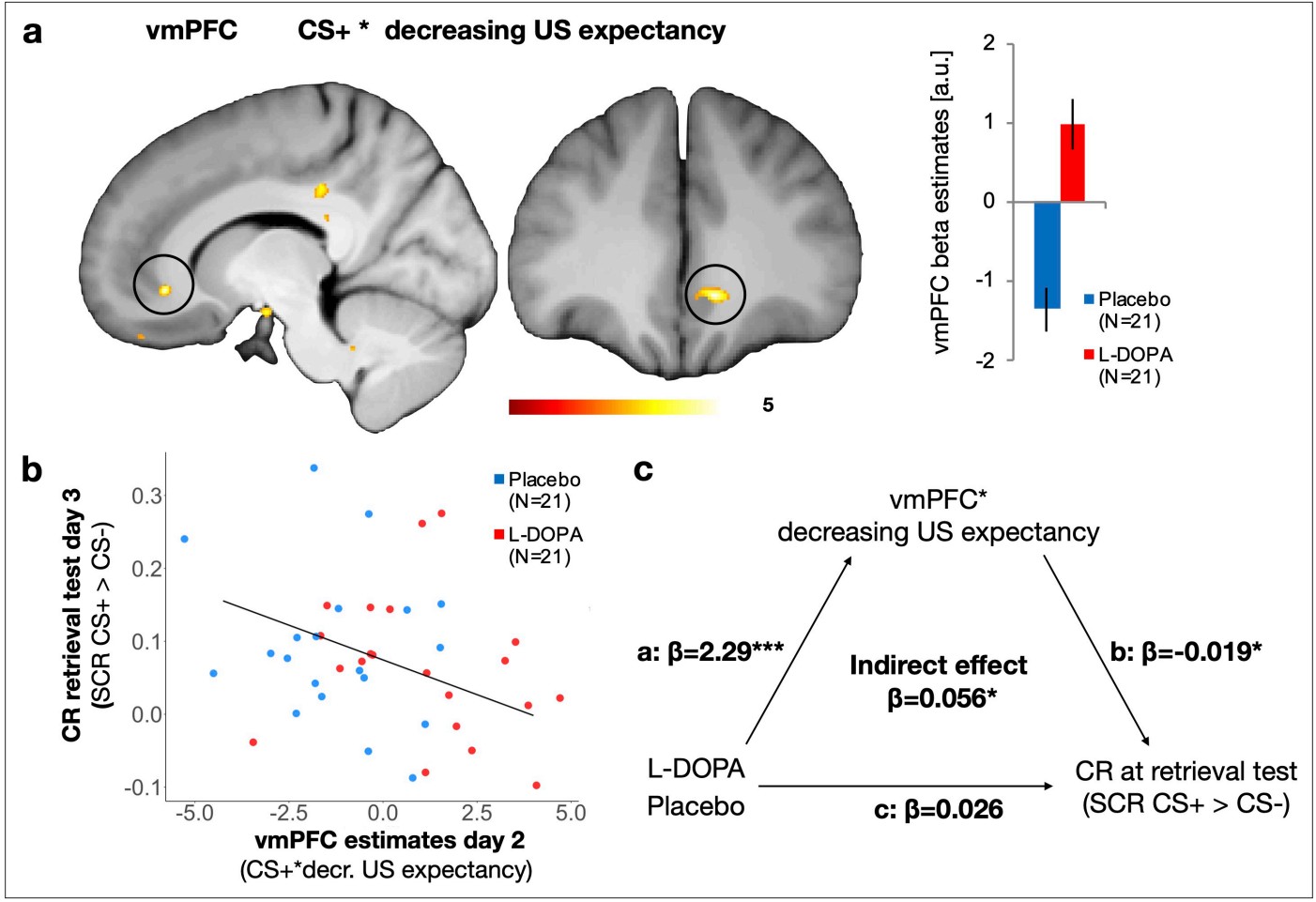

**Figure 3.** L-DOPA augmented vmPFC activity during extinction learning (day 2) that mediated retrieval of threat responses (day 3). (**a**) VmPFC responses that reflected CS+ trials during extinction training when subjects no longer expected an US that are contrasted with trials in which an US was expected (i.e., extinction learning). One-sided independent t-test L-DOPA > Placebo, MNI xyz: 10, 35,–7; Z = 4.76; $p_{FWE-SVC}$=0.002; displayed at threshold $p_{unc}$ <0.005; colour bar represents t-values. Estimates in the vmPFC were enhanced after administration of L-DOPA as compared to placebo (a.u. = arbitrary units; error-bar indicate the standard error of the mean). (**b**) Higher individual vmPFC responses across groups that reflected decreasing US expectancy for the CS+ (i.e., extinction learning) were associated with lower conditioned responses (SCR CS+ > CS−) during retrieval test 24 hr later (two-sided Pearson correlation). (**c**) The effect of L-DOPA treatment on conditioned responses (SCR CS+ > CS−) during retrieval test was fully mediated via the activity of the vmPFC in extinction learning. Drug treatment (L-DOPA vs. Placebo) had an effect on vmPFC activity (β = −2.2957, standard error = 0.4227, t(38)=-5.431, p=0.000003), and vmPFC activity had a negative effect on conditioned responses during retrieval test (β = −0.01898, standard error = 0.008888, t(38) = −2.135, p=0.0392). We found no evidence for an effect of drug treatment (L-DOPA vs. Placebo) on conditioned responses during retrieval test (β = 0.02644, standard error = 0.03238, t(38) = 0.816, p=0.419), but when including vmPFC activity into that model, this mediator was significant (β = −0.02478, standard error = 0.01192, t(38) = −2.079, p=0.0446; effect of group p=0.4). There was further evidence for a full mediation of drug treatment (L-DOPA vs. Placebo) on conditioned responses by an indirect effect of vmPFC activity within a mediation model using quasi-bayesian procedures (β = 0.0563, 95 % confidence intervals = 0.007–0.12, p=0.038, N = 40,1000 samples, N = 40); bootstrapping yielded comparable results (β = 0.056, 95 % confidence intervals = 0.007–0.14, p=0.044). a.u. = arbitrary units, vmPFC = ventromedial prefrontalcortex, CS = conditioned stimulus, US = unconditioned stimulus, SCR = skin conductance responses.

which was indirectly mediated by vmPFC activity during extinction learning (average causal mediation effect: β = 0.0563, 95 % confidence intervals = 0.007–0.12, p=0.038, quasi-Bayesian estimation of confidence intervals with 1000 iterations, N = 40, see *Figure 3* for detailed statistics).

As such, L-DOPA strengthens vmPFC activation that accompanies decreasing expectation of the US (i.e., extinction learning), and this enhancement of vmPFC activity mediates reduced differential SCR 24 hr later (i.e., better extinction memory retrieval). Our results thereby reveal an effect of L-DOPA on memory retrieval that is mediated by augmentation of vmPFC activity, which was specific for the individual time course of decreasing expectancy of the aversive outcome.

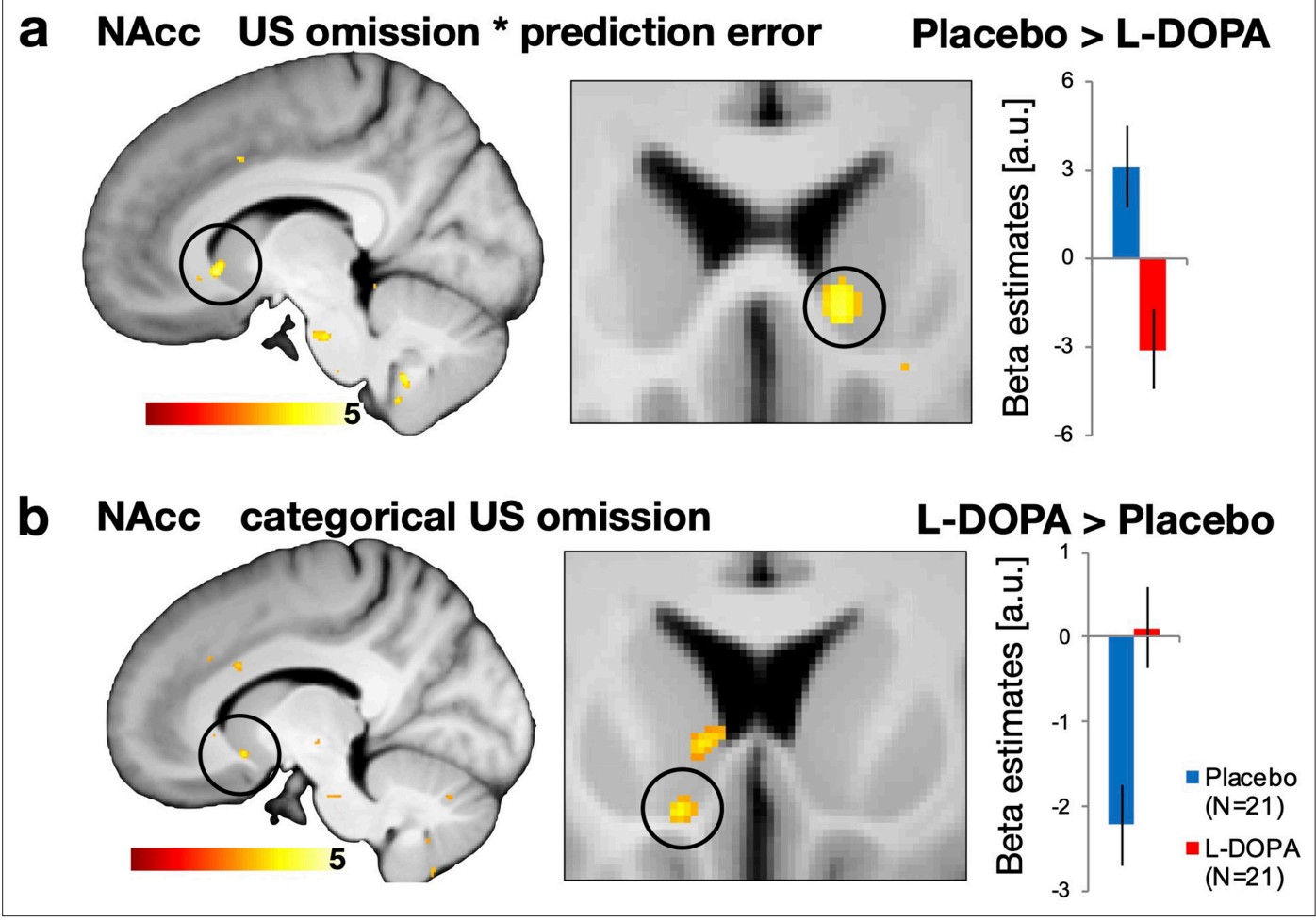

**Figure 4.** Omission of expected aversive outcomes in the NAcc is modulated by L-DOPA during extinction training on day 2. (**a**) At the time-point of US omission, the Placebo group exhibited expectancy violation coding (fitted prediction error term) in the right NAcc, which was not observed in participants that received L-DOPA (one-sided independent t-test Placebo > L -DOPA, MNI xyz: 9, 18,–4; $Z$ = 3.02; $p_{FWE-SVC}$=0.043). (**b**) Administration of L-DOPA abolished negative categorial responses (i.e., independent of expectancy) to omitted USs in the left NAcc that were found in Placebo controls (one-sided independent t-test L-DOPA > placebo, MNI xyz: –11, 14, –10; $Z$ = 3.22; $p_{FWE-SVC}$=0.029). Neural correlates are displayed at threshold $p_{unc}$ <0.005 with bar plot showing parameter estimates (a.u.). We found no group differences in functional connectivity during CS+ presentations in extinction training or acquisition training, as well as no differences during US presentations during acquisition training (see table *Supplementary file 1o*). a.u. = arbitrary units, NAcc = nucleus accumbens, US = unconditioned stimulus.

The online version of this article includes the following figure supplement(s) for figure 4:

**Figure supplement 1.** Enhanced associability related neural signalling in the left amygdala after the administration of L-DOPA.

## Omission of an expected aversive outcome is coded in the nucleus accumbens and modulated by L-DOPA

In the next step, we examined if decreasing US expectancy during extinction learning is driven by the omission of the US in form of an expectancy violation (i.e., prediction error) and if this process is modulated by dopamine. To this end, we used the modelled US expectancy ratings from the Rescorla-Wagner–Pearce-Hall hybrid model that has previously been used to describe computational processes in associative threat learning (*Boll et al., 2013*; *Li et al., 2011*; *Lindström et al., 2018*). In order to test for signals that reflect expectancy violation, we examined responses at the time-point of US omission that correlated with the modelled prediction error term (parametric modulator). We found that activation in the right nucleus accumbens in the Placebo group reflected the time course of the modelled prediction error term, but not in the L-DOPA group (see *Figure 4a*). This suggests that the nucleus accumbens in the Placebo group was responsive towards US omissions, only if an aversive outcome was still expected, which reflects the violation of the expected, yet omitted, value. This was supported

by an exploratory follow-up analysis, revealing a cluster in the right nucleus accumbens that reflected the expected value at the time-point of US omission (MNI xyz: 9, 18, –4; $Z$ = 3.04; $p_{FWE-SVC}$=0.040). Of course, prediction error and value are closely related, since the outcome during extinction training was the same for all trials. Next, we tested, if the L-DOPA group might show responses within the nucleus accumbens at the time-point of US omission that are independent of the prediction error (i.e., categorical responses). We found that categorical responses at the time-point of US were higher in the L-DOPA group, compared to Placebo group (see *Figure 4b*). In fact, responses in the Placebo group were negative, which would be in line with our finding of expectancy violation coding in the nucleus accumbens: Expectancy violation would be characterised by positive responses in early trials of extinction training (when US expectancy is high) and decreases rapidly with decreasing US expectancy, which could lead to negative responses when averaging a whole time course. Enhancing dopaminergic transmission (i.e., in the L-DOPA group), in contrast, seems to sustain responses to the omitted outcomes in the nucleus accumbens, irrespectively of expectancy of the US or value caching. Hence, our results imply a dopaminergic modulation of expectancy violation in the nucleus accumbens when expected aversive outcomes are omitted during extinction training. Note that within exploratory analyses that were suggested by the reviewers we found no activity in the nucleus accumbens to CS+ presentations during extinction (see *Supplementary file 1n*), which might suggest that the nucleus accumbens might rather be involved in processing of expectancy violation coding when expected USs are omitted. During acquisition training, however, we found activity in the nucleus accumbens to presentations of the CS+ (as well as in the contrast CS+ > CS−) and the US (see *Supplementary file 1n*), which resembles salience encoding, as reported in a recent study in rodents (*Cai et al., 2020*).

In addition to neural signalling that aligned with the prediction error term, we further investigated potential differences between groups in neural signals that follow the associability term, which provides a measure of prediction error-guided attention shift. Such attention-shifts denoted by associability involve several additional processes like arousal or awareness of the participants. We found that administration of L-DOPA enhanced associability related neural signals in the amygdala at the time-point of US omission, when compared to Placebo (see *Figure 4—figure supplement 1*). Our results suggest that dopaminergic enhancement might enhance shifting of attention or surprise that is initiated by unexpected omission of the US during extinction training.

## L-DOPA modulates functional connectivity between responses in the nucleus accumbens and the VTA when the US is omitted

Results in animals suggested that processes at the time-point of US omission involve not only the nucleus accumbens, but dopaminergic neurons in the VTA (*Salinas-Hernández et al., 2018*; *Cai et al., 2020*) and projections from the VTA to the nucleus accumbens (*Luo et al., 2018*), as well as projections from the basolateral complex of the amygdala to the nucleus accumbens (*Correia et al., 2016*).

To test whether the reported results in the nucleus accumbens at the time-point of the omitted US are functionally connected with other regions in the brain, we employed a condition-specific connectivity analysis with an anatomical nucleus accumbens (bilateral) mask as a seed region.

In line with animal data, we found stronger connectivity between the nucleus accumbens and the amygdala in the L-DOPA group when compared to the Placebo group (contrast: L-DOPA > Placebo, see *Figure 5a*). Moreover, contrasting responses at the time-point of US omission that were specifically improved by dopaminergic enhancement in the L-DOPA group, without making any assumption about the Placebo group (contrast: L-DOPA ≥ Placebo), revealed strengthened connectivity between the nucleus accumbens and the substantia nigra/VTA (SN/VTA) complex, see *Figure 5b*. Such coupling between the nucleus accumbens with the amygdala and the VTA was only observed to omitted USs during extinction learning, since additional control analyses did not conclusively support functional coupling between these regions during CS+ in extinction or acquisition training, as well as during US presentation in acquisition training (p(FEW) = 0.064; see table *Supplementary file 1o*). Hence, the activity in the nucleus accumbens during the omitted US is functionally coupled with responses in the amygdala and the SN/VTA and this connectivity is enhanced by administration of L-DOPA.

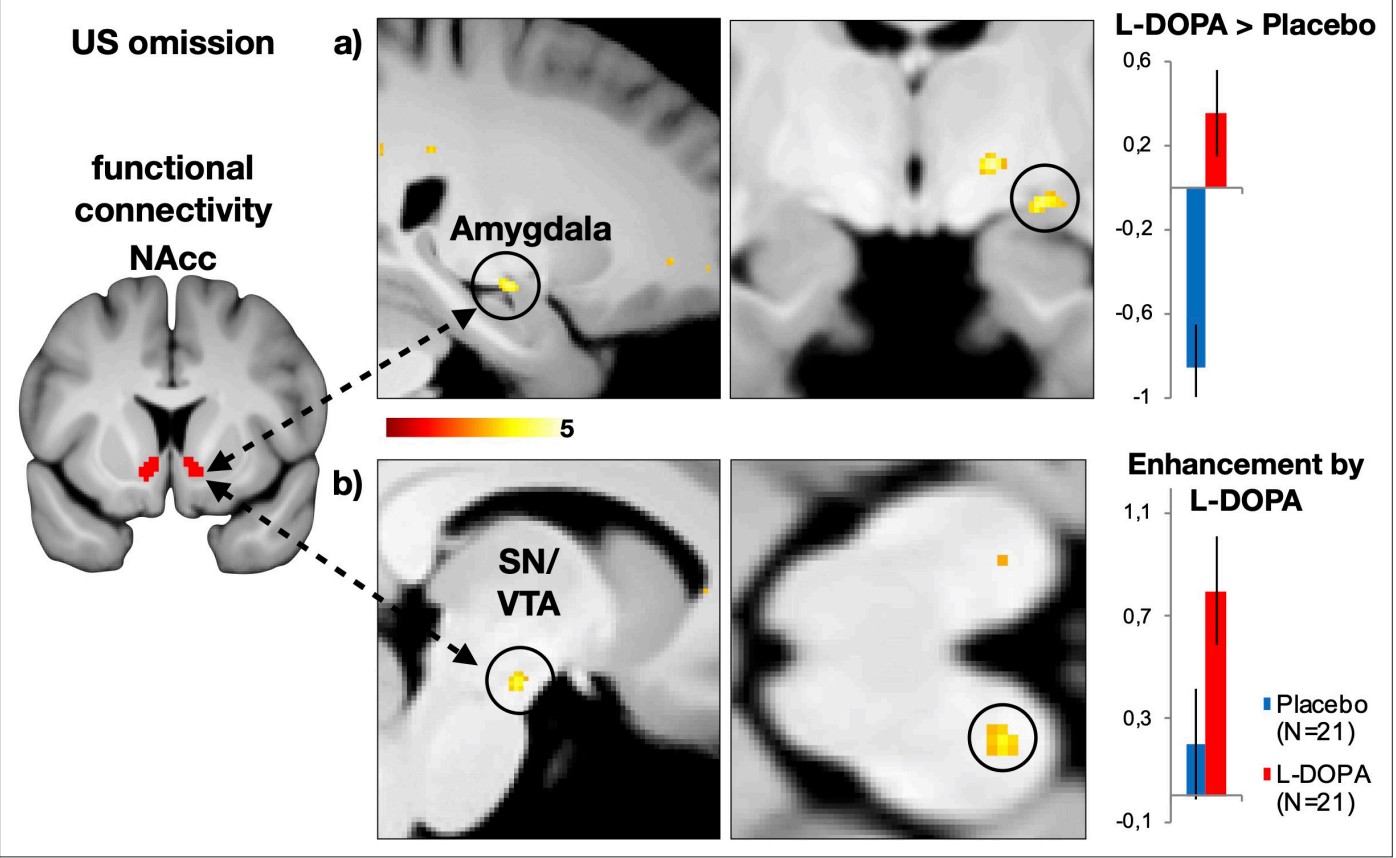

**Figure 5.** Dopaminergic modulation of functional connectivity between the nucleus accumbens and the VTA at the time-point of US omission during extinction training on day 2. (**a**) Administration of the dopaminergic precursor L-DOPA enhanced connectivity between the bilateral nucleus accumbens (seed, see left) and the left amygdala (one-sided comparison L-DOPA > placebo, MNI xzy: 24, –11, –10; Z = 3.73; p(FWE)=0.024), when compared to the Placebo group (contrast L-DOPA > Placebo). (**b**) Dopaminergic enhancement was furthermore associated with strengthened connectivity of the substantia nigra/ventral tegmental area complex and nucleus accumbens (one-sided comparison, L-DOPA = 1, Placebo = 0: MNI xzy: −8, –16,–15; Z = 3.43; p(FWE)=0.039). This contrast is specific to the enhancement of connectivity by L-DOPA, while making no assumptions about the Placebo group. The connectivity was condition-specific to the time-point of US omission (psycho-physiological interaction, PPI in SPM). We found functional connectivity during CS+ presentations in extinction training or acquisition training, as well as no differences during US presentations during acquisition training between these regions, as well as no differences between groups during CS+ presentations in extinction training (see **Supplementary file 1o**). T-maps are displayed at threshold $p_{unc}$ <0.005; colour bar represents t-values. a.u. = arbitrary units, NAcc = nucleus accumbens, SN/VTA = substantia nigra/ventral tegmental area complex.

## Discussion

Our results provide evidence that dopaminergic processes are involved in threat extinction learning. Dopaminergic enhancement during extinction learning augmented extinction memory at a later test, which was mediated by extinctio-learnin-specific vmPFC responses (i.e., reflecting decreasing US expectancy). Decreased US expectancy in extinction learning was further driven by dopaminergic activity within the nucleus accumbens that signalled the omission of expected aversive outcomes. This activity in the nucleus accumbens, when the US was omitted, was functionally coupled with the midbrain SN/VTA complex, as well as the amygdala. Additionally, we found reduced differential SCRs (CS+ > CS−, without strong support for a reduction in CS+ responses, only) in the L-DOPA group when compared with Placebo after the reinstatement one day after extinction learning. Hence, our results suggest that the activity of the nucleus accumbens, but not the vmPFC, encoded absent threats (prediction error) during extinction learning. Decreasing expectancy of threats in extinction learning was reflected by vmPFC activity, which was further enhanced by L-DOPA. These results corroborate findings from reinforcement learning in which outcome and prediction error signals were encoded in the striatum, whereas value was reflected by prefrontal regions (*Jocham et al., 2011*).

The main finding is that L-DOPA reduced retrieval of conditioned threat responses (SCR) by a mediation through enhanced vmPFC activity. In detail, we found that enhancement of dopaminergic transmission by the administration of L-DOPA (as compared to Placebo) during extinction learning enhanced individual neural signalling in the vmPFC that reflected the reduction of US expectation (i.e., extinction learning). These enhanced vmPFC responses were found to mediate the effect of L-DOPA on extinction memory retention, measured as reduced differential SCRs 24 hr later. Besides the implication of the vmPFC in safety signal processing (*Harrison et al., 2017*) and threat extinction in humans (*Kalisch et al., 2006*; *Milad et al., 2007*; *Phelps et al., 2004*; *Ball et al., 2017*; *Merz et al., 2018*), our results align specifically with a previous finding of vmPFC activity pattern during extinction memory consolidation that mediates the effect of L-DOPA on extinction memory retention (*Gerlicher et al., 2018*). Our results extend this finding on memory consolidation by providing a link between dopaminergic effects on vmPFC activity that are specific for individual extinction *learning* (i.e., decreasing US expectancy) and augmentation of extinction *memory*. L-DOPA might have the potential to improve (otherwise low, see parameter estimates in the Placebo group, *Figure 3b*) vmPFC activity during extinction learning. Hence, rather than enhancing extinction learning per se, L-DOPA administration seems to augment vmPFC responses that accompany decreased US expectancy. This suggests a benefit of L-DOPA for extinction *learning* processes. Our findings would fit to previous results that link the benefit of neuropharmacological intervention in extinction learning to decreasing threat expectancies (*Sartori and Singewald, 2019*; *Gerlicher et al., 2019*; *Hofmann, 2014*; *Smits et al., 2013*). Thereby, L-DOPA might have the potential for a psychopharmacological treatment that augments threat extinction learning instead of dampening overall threat responses, like classic anxiolytics. Additionally, we found that L-DOPA administration during extinction training reduced differential SCRs after reinstatement, which aligns with a finding of decreased SCRs after reinstatement by L-DOPA administration that followed extinction training in women diagnosed with post-traumatic stress disorder (*Cisler et al., 2020*).

A second set of our results implicate that decrement of US expectancy in extinction training involves a dopaminergic coding of expectancy violation in form of a prediction error at the time-point of US omission. In detail, administration of L-DOPA enhanced general (categorical) activity during omitted US, which was paralleled by reduced prediction error signals that we found in the placebo group. The enhanced general activity in the nucleus accumbens to omitted outcomes might results from enhanced activity in dopaminergic projections from midbrain, which would be in line with L-DOPA-induced dopamine release in the midbrain in humans at rest (*Black et al., 2015*). L-DOPA has further been found to activate otherwise silent dopaminergic neurons in animals (*Grace, 2008*), which thereby might enhance activity in the nucleus accumbens in general. In parallel, L-DOPA could have induced inhibitory signals (via dopamine D2-receptors) that limit phasic burst-firing and thereby reduce the pattern of prediction error encoding (*Beckstead et al., 2004*). Such a shift in striatal activity by L-DOPA would fit to the observation of an opposite effect by blockade of dopamine D2 receptors (using haloperidol [*Wächtler et al., 2020*]): reduced categorical activity during (rewarding) outcomes and enhanced prediction error signals. Our findings furthermore align with a negative association between striatal dopamine synthesis capacity (measured by radiolabelled L-DOPA during positron emission tomography) and BOLD responses related to reward prediction errors (*Schlagenhauf et al., 2013*). Hence, higher dopamine synthesis capacity (indicated by uptake of L-DOPA in the striatum) was associated with lower prediction error related BOLD activity.

Our study was intentionally not designed to disentangle details of expectancy violation coding of omitted USs, but rather to provide a scenario of safety learning. While we cannot disentangle encoding of salience and expectancy violation in the nucleus accumbens within our experiment, we conducted additional analyses in order to differentiate activity to the omitted US from other salient events like the CS+ or the US. During extinction training, we found no involvement of the nucleus accumbens during presentations of the CS+, which is more in line with coding of expectancy violation rather than salience (see *Supplementary file 1n*). During acquisition training, however, the nucleus accumbens was associated with salience encoding, since we found activation to presentations of the CS+ (and in the contrast CS+ > CS−), as well as the US (see *Supplementary file 1n*). Additionally, we found during acquisition training a connectivity between the nucleus accumbens and the VTA during CS+ presentations, albeit less statistical support (p(FWE) = 0.064, see *Supplementary file 1o*). A recent study in rodents could disentangle encoding of salience and expectancy violation

during extinction learning in lateral and medial part of the VTA, respectively (*Cai et al., 2020*). While we cannot examine such fine-grained processes in the VTA with fMRI in humans, we nevertheless provide evidence for a role of the nucleus accumbens in processing of US omissions, which is in line with a function of the nucleus accumbens in rodents (*Badrinarayan et al., 2012*; *Oleson et al., 2012*; *Budygin et al., 2012*) in particular during extinction learning (*Correia et al., 2016*; *Holtzman-Assif et al., 2010*; *Rodriguez-Romaguera et al., 2012*; *Whittle et al., 2013*). Our results furthermore dovetail with two previous neuroimaging study reporting prediction error signals in the nucleus accumbens during extinction training (*Thiele et al., 2021*; *Raczka et al., 2011*). Our results point moreover to a dopaminergic modulation of surprise in the amygdala that is evoked by US omission, which fits well to previous reports of associability coding in the amygdala during threat learning (*Boll et al., 2013*; *Lindström et al., 2018*).

We further show that signals in nucleus accumbens at the time-point of US omission were functionally coupled with activation in the amygdala and the SN/VTA, which were enhanced by administration of the dopaminergic precursor L-DOPA. This finding mirrors findings in animals implying neurons in the VTA, as well as projections from the VTA to the nucleus accumbens, in the encoding of the omission of an expected US (*Luo et al., 2018*; *Salinas-Hernández et al., 2018*). Furthermore, our results would align with studies in animals that provided evidence for amygdala to nucleus accumbens projections that underlie extinction of threat responses (*Correia et al., 2016*). Such interaction between dopaminergic pathways and amygdala activity during threat extinction is moreover in line with a study in rodents that revealed changes in inhibitory interneuron activity in the amygdala by dopaminergic projections from the VTA that enable suppression of freezing during early extinction (*Aksoy-Aksel et al., 2021*).

## Limitations of this study

This pharmacological fMRI study in human volunteers is only suited to draw inferences on blood-oxygen-level-dependent signals as a function of L-DOPA administration. Hence, the changes reported here are only indirect markers of changes in dopaminergic neurotransmission, which might suggest together with fine-grained studies in animals that dopaminergic neurotransmission is involved in extinction learning across species (*Felsenberg et al., 2018*; *Luo et al., 2018*). Dopamine-specific markers in humans (e.g., using PET) might be suitable to unambiguously link changes in vmPFC and nucleus accumbens activity during extinction learning to dopaminergic neurotransmission.

The behavioural effects of pre-extinction administration of L-DOPA in our study were weaker when compared with studies that employed post-extinction administration (*Gerlicher et al., 2018*; *Haaker et al., 2015*; *Haaker et al., 2013*; *Gerlicher et al., 2019*). We found no support in univariate analyses that L-DOPA administration decreased conditioned responding across all outcome measures that reflect different threat processing, such as US expectancy, psycho-physiological arousal (SCR), and affective (fear) ratings during extinction learning and retrieval test. Future studies might employ more fine-grained explicit ratings of threat expectancy or use other psycho-physiological such as pupil-size or startle responses (*Tzovara et al., 2018*; *Korn et al., 2017*).

Nevertheless, we found that L-DOPA decreased differential SCRs during retrieval test by mediation via vmPFC activity during extinction learning, which converges with a previous finding (*Gerlicher et al., 2018*). Hence, the current state of research might suggest that L-DOPA enhances vmPFC activity during or after extinction training, that could decrease threat responses, rather than blunting threat responses, per se.

Our current sample size was based on effect sizes that result from previous reports of post-extinction administration (*Gerlicher et al., 2018*; *Haaker et al., 2013*), and hence our sample size might have been too small to detect univariate differences between the L-DOPA and placebo group with sufficient power.

Our results were furthermore derived from a population of healthy male volunteers and a role for L-DOPA in extinction learning within a representative sample, as well as populations with anxiety disorders have just been recently pioneered (*Cisler et al., 2020*). Future studies that further investigate L-DOPA as a novel augmentation strategy for the therapy of anxiety-related disorders (in which extinction mechanisms are only one part of the process) are warranted.

In sum, our results thereby provide a neuropharmacological, dopaminergic mechanism for augmentation of neural substrates that underlie extinction learning in humans (*Gerlicher et al., 2019*;

*Hofmann, 2014*; *Smits et al., 2013*; *Smits et al., 2013*), which could provide a promising novel strategy to augment behavioural treatments of anxiety related disorders (*Cisler et al., 2020*).

## Materials and methods
### Participants
Fifty healthy male subjects without self-reported psychiatric and neurological diseases, without current medication (including no 'over the counter drugs' without prescription within the last 2 weeks), or current use of illicit drugs (urine toxicology) were recruited in this study. Illegal drug-screening test was carried out prior to testing at day 1 (M-10/3-DT; Diagnostik Nord). The final sample in the analyses included 46 participants (L-DOPA N = 24, Placebo N = 22) between the age of 20 and 38 (mean 27.07, SD = 4.18; L-DOPA mean = 27.29, SD = 4.102; Placebo mean = 26.82, SD = 4.35; two-sided unpaired t-test: t(44)=0.38, p>0.7) after exclusion of four subjects (positive drug urine test N = 1, incidental finding of a brain cyst N = 1, not following the instructions N = 1 and accidental press of the emergency bell N = 1).

The sample size of 40 participants (plus 10 drop-outs) was determined a priory in order to archive a power of 0.95 with an alpha level of 0.05 and assuming an effect size of eta$^2$ = 0.08 (previous effect of L-DOPA on extinction memory consolidation, G*Power 3.1.9.6, RRID:SCR_013726).

The study (including sample size approximation) was approved by local ethics committee in Hamburg (Ärztekammer Hamburg). Full participation of this study was remunerated with 120,- EURO.

### Stimulus material
#### Conditioned stimuli
Contexts surrounding the CSs were employed as computerised environments of virtual offices (Source Engine, Valve Corporation, Bellevue, WA, used in *Andreatta et al., 2015*). Each office image was depicted from two different vantage points (on the wall opposite the door vs. on the wall to the right of the door). Three different contexts were used: context A, context B, and a mixture of both in order to induce a contextual generalization (*Andreatta et al., 2015*). The context that included a mixture of contexts A and B contained 50 % of the furniture from context A and 50 % from context B, equally distributed in the room. Virtual offices consisted of the same floor plan but differed regarding the furniture. A blue or a yellow color filter illuminating the whole room (duration of 6 s) served as CSs, indicating either CS+ or CS−. Colours of the CSs and contextual backgrounds were counterbalanced across participants. Presentation of the context served as the inter-trial intervals (ITIs, duration range 7–11 s, mean 7.8). The visual stimulus material was presented in pseudo-randomized order on a computer screen using Presentation software (NeuroBehavioral Systems, Albany, CA; RRID:SCR_002521).

#### Unconditioned stimulus
An electrotactile stimulus consisting of a train of three square-wave pulses of 2 ms duration each (interval 50 ms) served as the US that the CS+ onset after 5 s. The US was delivered through a surface electrode with platinum pin (Specialty Developments, Bexley, UK) on the right dorsal hand using a DS7A electrical stimulator (Digitimer, Welwyn Garden City, UK). US intensity was individually adjusted prior to acquisition training (day 1) to a level of maximal tolerable pain (mean 8.1 ± 0.5 mA, range 2.5–21.0 mA), and participants were asked to rate the aversiveness of the US between 0 ('I feel nothing') and 10 ('maximally unpleasant'; rating: mean 7.1 ± 0.1, range 4.0–8.0). Additional US intensity ratings were acquired after fear acquisition training (between 0 and 100 day 1: mean 68.65 ± 3.0, range 20–100) and at the end of return of fear testing (day 3: mean 49.91 ± 3.9, range 0–100). There were no differences between the Placebo and the L-DOPA group in any of these parameters (all p>0.167; see *Supplementary file 1a*).

### Study medication
Study medication included an oral administration of 150 mg L-DOPA (including 37.5 mg benserazide) in a double-blind and placebo-controlled protocol 60 min before extinction training. Participants were allocated into the placebo or L-DOPA group before day 1 in a restricted randomisation procedure that allocated five subjects to the L-DOPA and five subjects to the placebo group for each group

of 10 participants. The dose of 150 mg has been found effective in previous studies to enhance the consolidation of extinction memories in humans (*Gerlicher et al., 2018*; *Haaker et al., 2015*; *Haaker et al., 2013*).

## Experimental procedure

Using a three-day paradigm, acquisition training (day 1) and extinction training (day 2, approx. 24 hr after acquisition) were conducted in the fMRI scanner, while retrieval test (day 3, approx. 24 hr after extinction), including reinstatement, were employed within the psycho-physiological laboratory. Acquisition training took place in context A, whereas extinction training was employed in context B. Retrieval test (including reinstatement procedure) was conducted in a 50/50-mixture of context A and B in order to examine contextual generalisation (*Andreatta et al., 2015*), which also involves contextual renewal of conditioned threat responses (*Vervliet et al., 2013*). Twenty-four hours after acquisition training participants received L-DOPA (see Study medication) before the extinction learning session (the CS+ was no longer followed by an aversive outcome). L-DOPA administration thereby affected extinction training, while acquisition training, as well as retention and reinstatement tests were conducted drug free. Data collection on day 1 included sampling of plasma concentration of endocannabinoids as part of a different project.

### Acquisition training (day 1)

A short habituation phase preceded acquisition training (six trials: three CS+, three CS−) without any presentation of the US. Subsequent acquisition training consisted of 24 trials for each CS (in context A). The CS+ was followed by a US in 75 % of the trials, whereas the CS− was never followed by a US. Participants were not informed about the conditioning contingencies or the learning element beforehand.

### Extinction training (day 2)

Approximately 24 hr after conditioning, participants returned to the fMRI laboratory. US and SCR electrodes were attached exactly as the day before, without US intensity adjustment. During extinction training, 24 trials (context B) were presented for each CS, and no US was administered. Participants were not informed beforehand about any change in CS–US contingencies.

### Retrieval test and reinstatement (day 3)

Participants returned to the psycho-physiological laboratory and US and SCR electrodes were again attached without further US adjustment. A retrieval test (contextual generalisation in a 50/50-mixture of context A and B, i.e. one context that 50% of the furniture from context A and 50% from context B) consisted of eight unreinforced trials of each CS and was followed by four unsignalled reinstatement USs (interval duration range 10–15 s). Here, the same individual electrical stimulation intensity was used as during acquisition training. Six to 10 s after the last reinstatement US, a second retrieval test (reinstatement test) was employed, including 16 trials (with no US) of each CS. The order of CS+ and CS− after the reinstatement US was counterbalanced across subjects. At the end of the experiment, CS–US contingency awareness was assessed using a semi-structured interview (*Bechara et al., 1995*) and based on these results 37 participants were classified as aware and five were classified as unaware (no differences between groups, $\chi^2$-test, p=0.634).

## Outcome measures and analyses

### US expectancy

On each CS trial presentation, participants had to rate their US expectancy as a binary choice (key press for yes/no) without any scale presented to avoid any distraction. Participants were excluded from the analyses (day-wise) if less than one third of all data points were missing (excluded participants: N[day 1] = 0, N[day 2] = 3, N[day 3] = 3).

### Fear ratings

At the beginning as well as at the end of each experimental day, participants were asked to rate the fear/stress/tension level that was elicited by each CS. On day 1, the first rating was conducted after

habituation phase and before acquisition training. Ratings were performed on a computerised Visual Analogue Scale (VAS, 0 [none] – 100 [maximal]) using keys with the right hand. Rating values had to be confirmed by a key press (otherwise missing data, N[day 1] = 0, N[day 2] = 3, N[day 3] = 4). All rating values were range-corrected (divided by the maximal rating value on that day).

## Skin conductance

Skin conductance responses (SCRs) were measured via self-adhesive Ag/AgCl electrodes placed on the palmar side of the left hand on the distal and proximal hypothenar. Data were recorded with a BIOPAC MP-100 amplifier (BIOPAC Systems Inc, Goleta, CA; RRID:SCR_014829) using AcqKnowledge four software (RRID:SCR_014279). For data analysis, SCR signal was down-sampled to 10 Hz and responses were manually scored between 0.9 and 4.0 s after CS onset using a custom-made computer program. Non-reactions were scored as zero, and trials with obvious electrode artefacts were scored as missing data. Afterwards, amplitudes were logarithmised and range-corrected (SCR/SCRmax CS [day]) separately for the three consecutive experimental days in order to account for inter-individual variability. SCR data from a limited number of participants had insufficient data quality (as judged by two researches; due to signal-disturbances by the fMRI acquisition) and were thus excluded (day-wise) before the analyses (N[day 1] = 1, N[day 2] = 6, N[day 3] = 4). Trial-wise SCRs were then averaged over a block of eight trials, resulting in three blocks on each day.

## Statistical analysis

Outcome measures were analysed (using JASP 0.11.1, JASP Team (2020) [Computer software], RRID:SCR_015823) employing repeated-measures ANOVAs. For acquisition and extinction training, these ANOVAS included CS-type (2) and the effect of time (fear ratings: 2 ratings, SCR and US expectancy: 3 blocks, each average across eight trials). Pharmacological group was entered as a between subject factor. For day 3, we analysed the first block separately as the retrieval test and the reinstatement analyses included two comparisons of trials before and after the reinstatement USs (*Haaker et al., 2014*). First, we compared responses averaged across the whole block (eight trials) before and after reinstatement. Since reinstatement effect are transient and only detectable over a few trials, we added a second, more detailed analysis, which compared responses averaged across the three trials before and after reinstatement, based on previous findings indicating that transient reinstatement effects can be found up to three trials after the US presentation (*Scharfenort and Lonsdorf, 2016*). In all analyses, an α-level of p<0.05 was adopted and sphericity correction (Greenhouse-Geisser) was applied. Follow-up post hoc test on measurement on days 2 and 3 was performed as one-sided independent t-test to examine the hypothesis of L-DOPA responses < Placebo responses. During data acquisition, preprocessing and initial analyses, the experimenter were masked to the drug conditions.

## Hybrid model

To examine how decreasing US expectancy is driven by expectancy violation from the omission of the US, we fitted trial-wise US expectancy ratings with a Rescorla-Wagner–Pearce-Hall hybrid model, which is the same model employed in previous neurocomputational studies of aversive learning in humans (*Boll et al., 2013*; *Li et al., 2011*).

In order to examine associative threat learning processes, which can be described by classical formal learning theory such as the Rescorla-Wagner (R-W) (*Rescorla et al., 1972*) and Pearce-Hall (P-H) model (*Pearce and Bouton, 2001*), we analysed extinction learning underlying mechanisms based on trial-by-trial US expectancy ratings. Therefore, a Rescorla-Wagner–Pearce-Hall hybrid model (HM) (*Le Pelley, 2004*) was used, which algebraically describes error-driven learning based on prediction errors (PE, i.e., mismatches) between the predicted (aversive) outcomes (denoted as expected "values," $v$) and the received outcomes (RO), which in this case corresponded to the omissions of the US. Extending the RW model, the HM explicitly accounts for dynamically changing learning rates $\alpha$ (i.e., surprising absence of the US) that is updated depending on the associability $\eta$ (i.e., the reliability of prior predictions). That means the associability $\eta$ increases in proportion to the absolute prediction error (PE) on the last interaction with a stimulus, allowing the agent to adapt to changing environments, which leads to larger prediction errors (PE), and thereby higher associability $\eta$. The HM is formalised by the following equation:

$$v_{t+1} = v_t + \alpha * \eta_t * PE$$

The predicted 'values' ($v$) on the next trial $t + 1$ are based on the 'value' at the current trial $t$ and on prediction errors (PE) scaled by the learning rate $\alpha$ and the current associability $\eta_t$. Prediction errors (PE) are calculated as the difference between the current predicted values ($v_t$) and the received outcomes (RO).

$$PE = RO - v_t$$

The current associability $\eta_t$ is updated according to the absolute prediction error (PE) and the associability of the preceding trial $\eta_{t-1}$ with the free scaling parameter $\omega$.

$$\eta_t = \omega * \big|(PE)\big| + \big(1 - \omega\big) * \eta_{t-1}$$

The model employs a softmax function with a free 'inverse temperature' parameter $\beta$ to generate trial-by-trial probabilities ($p$) for the binary US expectancy ratings.

$$p = \frac{1}{1 + e^{-v/\beta}}$$

The model thus contains three free parameters: (1) the learning rate $\alpha$, (2) the scaling parameter $\omega$ for the associability $\eta$, and (3) the inverse temperature parameter $\beta$. These three free parameters were initialised in the fitting procedure as 0.5, 0.5, and 4, respectively. The starting point for the initial 'value' $v_0$ was set to 0.75, i.e., the probability for a US following a CS+ in the acquisition phase. The starting point for the initial associability $\eta_0$ was set to 1, which assumes that the associability is initially fully dependent on the prediction error (PE). We fitted model parameters using maximum likelihood estimation (MLE). Specifically, we used the non-linear Nelder-Mead simplex search algorithm (implemented in the MATLAB [RRID:SCR_001622] function fminsearch) to minimise negative log-likelihood summed over all trials for each participant.

## fMRI acquisition and analysis

MRI data were obtained on a 3T Magnetom-PRISMA System (Siemens, Erlangen, Germany) using a 64-channel head coil. fMRI measurements were performed using single-shot echo-planarimaging with parallel imaging (GRAPPA, in-plane acceleration factor 2) (*Griswold et al., 2002*) and simultaneous multi-slice acquisitions ('multiband', slice acceleration factor 2) (*Feinberg et al., 2010*; *Feinberg et al., 2010*; *Moeller et al., 2010*) as described in *Setsompop et al., 2012*. The corresponding image reconstruction algorithm was provided by the University of Minnesota Center for Magnetic Resonance Research. Echo planar multiband images were acquired with 42 continuous axial slices (1.5 mm thickness, 0.5 mm gap) in a T2*-sensitive sequence (TR = 1493 ms, TE = 30 ms, flip angle = 60°, field of view = 225 × 225 mm²). Selection of slice arrangement was individually adjusted (to the dorsal anterior cingulate cortex as an orienting point) in order to cover the following areas: ventral medial prefrontal cortex, nucleus accumbens, amygdala, and midbrain SN/VTA. Moreover, high-resolution T1-weighted structural brain image (MP-RAGE sequence, 1 mm isotropic voxel size, 240 slices) were obtained. For task-relevant functional data of day 2 (extinction training), preprocessing and statistical analysis was carried out using SPM12 (Statistical Parametric Mapping, http://www.fil.ion.ucl.ac.uk/spm, RRID:SCR_007037) running under Matlab2017a (The MathWorks, Inc, Natick, MA). To account for T1 equilibrium effects, the first five volumes of each time-series were discarded. All remaining images were unwarped, realigned to the first image, coregistered to the individual high-resolution T1 structural image. Subsequent statistical analyses were performed by using a standard approach for fMRI implemented in the SPM software, involving a general linear convolution model (GLM) at the single-subject level and a random-effects analysis on group level. On individual-level, experimental conditions (i.e., ITI, CS+, CS−, omitted US, introductions, ratings, and button presses) were defined as separate regressors modelling the predicted time courses of experimentally induced brain activation changes as a stick function. Furthermore, CS+ regressors included a parametric modulation of individual US expectancy ratings in order to examine dopamine-dependent differences in neural representation in decreasing US expectancy during extinction learning. Additionally, parametrical modulation of the omitted US was applied to examine neural responses that are related to changes in expectancy violation over trials. Therefore, the modelled prediction error term (as a measure of

expectancy violation, averaged across the whole sample) and the orthogonalised associability term (as a measure of prediction error-guided surprise, averaged across the whole sample) were entered trial-wise.

In a next step, subject- and regressor-specific parameter estimate images of interest were normalised to a sample-customised DARTEL template (*Ashburner, 2007*) smoothed with an isotropic full-width at half-maximum Gaussian kernel of 4 mm. These estimates were then included into separate random-effects group analysis using SPM's 'full factorial' model, which permits correction for possible non-sphericity of the error term (here, dependence of conditions). Model factors for the respective analysis were CS+*US expectancy ratings (extinction learning), omitted US*mean prediction error and omitted US*associability (expectancy violation), always including the factor group (Placebo, L-DOPA). Analyses main objective was to reveal dopamine-specific effects (L-DOPA vs Placebo) during extinction learning depending on time-point related changes in US expectation. Significance of effects was tested by using voxel-wise one-tailed t tests. According to our hypotheses, we expected enhanced signalling in the L-DOPA group as compared to the Placebo group in CS+*US expectancy ratings (extinction learning) in the vmPFC (L-DOPA > Placebo). We further expected a modulation of omitted US signals by L-DOPA administration and examined all contrast for the time-point of US omission (omitted US*mean prediction error, omitted US*associability, categorical omitted US) in both directions (L-DOPA > Placebo and Placebo > L -DOPA).

Regions of interest (ROI) were the defined as (1) dopaminergic key structures, such as the nucleus accumbens and the VTA/SN and (2) key structures in extinction learning, such as the amygdala and the vmPFC. These structures were defined by Havard-Oxford probability maps for the nucleus accumbens and the amygdala (*Desikan et al., 2006*). For the SN/VTA and vmPFC is no anatomical mask available, therefore we defined both ROIs as in a previous study that revealed an effect of L-DOPA treatment in both, the SN/VTA and vmPFC ROI (*Lonsdorf et al., 2014*). The SN/VTA complex was defined by *Bunzeck and Düzel, 2006*. The vmPFC ROI was defined as a box of 20 × 16 × 16 mm at x = 0, y = 42, z = −12. Correction for multiple comparisons within these ROIs was performed by using family-wise error correction based on the Gaussian Random Fields as implemented in SPM.

## Connectivity analysis

Psycho-physiological interaction (PPI, as implemented in SPM12) was used to examine functional connectivity differences of responses in the nucleus accumbens towards the omitted US between groups. Extracted eigenvariates of nucleus accumbens (bilateral ROI mask) were used as the seed region, deconvolved and multiplied with the condition specific onsets of the omitted US. The product (PPI) was entered as a regressor into an individual GLM for each participant, controlling for the time course of the nucleus accumbens, the onset regressor, and movement as nuisance regressors. Parameter estimates of the omitted US-PPI were then contrasted between groups.

## Mediation analysis

To test whether the effect of L-DOPA vs Placebo on differential SCRs at retrieval test on day 3 was mediated by the activity in the vmPFC that aligned with decreasing US expectancy, we employed a mediation analysis (R Studio, Version 1.2.1335, package 'mediation'; R Project for statistical computing RRID:SCR_1905). This analysis was based on a prior analysis that revealed that effects of L-DOPA on extinction memory retention (differential SCRs during retention test) were mediated by vmPFC activity during consolidation' (*Gerlicher et al., 2018*).

## Acknowledgements

The authors thank Smilla Weisser for support with the recruitment of participants and Katrin Bergholz, Kathrin Wendt, and Waldemar Schwarz for help with MR-data acquisition. We further thank Jürgen Finsterbusch for the development of the MRI sequence, and the authors are grateful to the University of Minnesota Center for Magnetic Resonance Research for providing the image reconstruction algorithm for the simultaneous multi-slice acquisitions. Funding/Support: RE and JH were supported by the Collaborative Research Center TRR58 'Fear, Anxiety, Anxiety Disorders', Projectnumber 44541416, Project B10 (INST 211/755), funded by the German Research Foundation (DFG). JH was further supported by an individual research grant 7470/3-1 from the German Research Foundation

(DFG). CWK was supported by two grants from the German Research Foundation (DFG): the collaborative research centre SFB TRR 169 and an Emmy Noether Research Group (392443797).

## Additional information

### Funding

| Funder | Grant reference number | Author |
|---|---|---|
| Deutsche Forschungsgemeinschaft | Project B10 (INST 211/755) of the Collaborative Research Center TRR58 | Roland Esser Jan Haaker |
| Deutsche Forschungsgemeinschaft | HA 7470/3-1 | Jan Haaker |
| Deutsche Forschungsgemeinschaft | collaborative research centre SFB TRR 169 | Christoph W Korn |
| Deutsche Forschungsgemeinschaft | Emmy Noether Research Group (392443797) | Christoph W Korn |

The funders had no role in study design, data collection and interpretation, or the decision to submit the work for publication.

### Author contributions

Roland Esser, Data curation, Formal analysis, Investigation, Project administration, Validation, Visualization, Writing - original draft; Christoph W Korn, Formal analysis, Methodology, Resources, Writing - original draft, Writing – review and editing; Florian Ganzer, Investigation, Project administration, Writing – review and editing; Jan Haaker, Conceptualization, Formal analysis, Funding acquisition, Project administration, Supervision, Writing - original draft, Writing – review and editing

### Author ORCIDs

Christoph W Korn (iD) http://orcid.org/0000-0002-8228-2624
Jan Haaker (iD) http://orcid.org/0000-0001-8366-9559

### Ethics

Human subjects: The ethical approval was obtained by the ethics committee of the Ärztekammer Hamburg (PV5158) that approved the study. Participants gave their written, informed consent to participate in the study, for the collection of the data and consent to publish.

### Decision letter and Author response

Decision letter https://doi.org/10.7554/eLife.65280.sa1
Author response https://doi.org/10.7554/eLife.65280.sa2

## Additional files

### Supplementary files
- Transparent reporting form
- Supplementary file 1. Supplementary materials.

### Data availability

All data for analyses and figures in this study are provided in the within the Open Science Framework.

The following dataset was generated:

| Author(s) | Year | Dataset title | Dataset URL | Database and Identifier |
|---|---|---|---|---|
| Haaker I | 2020 | Dopamine_extinction_learning_esser_et_al | https://osf.io/6tfu3/?view_only=50493b0ad31f4801953873363f9f9ec2 | Open Science Framework, 10.17605/OSF.IO/6TFU3 |

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
