## [Decision Letter]

**Acceptance summary:**

In the present study, the authors examined the impact of the dopamine precursor, L-DOPA, on the extinction of experimental fear. L-DOPA, or placebo was administered to fear conditioned healthy volunteers and fMRI was used to quantify brain activity during extinction training. Behavioral rating and skin-conductance responses serve to validate fear acquisition, extinction learning, and extinction retrieval. Results indicated that L-DOPA improved fear extinction as indexed by lower fear responses during tests for fear relapse, and this was associated with enhanced activity in the ventromedial prefrontal cortex.

**Decision letter after peer review:**

Thank you for submitting your article "Dopaminergic signals in the Nucleus Accumbens, VTA and vmPFC underpin extinction learning from omitted threats" for consideration by *eLife*. Your article has been reviewed by 3 peer reviewers, and the evaluation has been overseen by Dr. Shackman as the Reviewing Editor and Dr. Frank as the Senior Editor. The reviewers have opted to remain anonymous.

Summary

In the present study, the authors examined the impact of L-DOPA on the extinction of experimental fear. The dopamine precursor, L-DOPA, or placebo was administered to fear conditioned healthy volunteers and fMRI was used to quantify brain activity during extinction training. Behavioral rating and skin-conductance responses serve to validate fear acquisition, extinction learning, and extinction retrieval. Results indicated that L-DOPA improved fear extinction as indexed by lower fear responses during tests for fear relapse, and this was associated with enhanced activity in the ventromedial prefrontal cortex. Authors also observed increased functional connectivity with SN/VTA and amygdala.

The reviewers expressed some enthusiasm for the manuscript:

– The study is of interest to a broad community and may have as well impact on societies by improving exposure-based psychotherapies for the treatment of anxiety disorder.

– This paper will be of high interest for neuroscientists, both human and animal researchers, in particular in the area of learning and memory and its translational –clinical application.

– The manuscript is well-written and addresses an important and timely topic with clinical relevance and hence is of general interest to the scientific community.

– These are interesting and important questions to be dealt with in human studies, and important for basic conditioning research and application (e.g. suggesting the use as an adjunct of pharmacological treatment in behavioral therapy of conditioning based anxiety disorders).

Essential revisions:

Nevertheless, all 3 reviewers expressed some significant concerns.

• The reviewers highlighted some major concerns regarding the framing and interpretation of the results (as described below, and in the specific comments on the Results/Discussion, below)

o Title – The title "Dopaminergic signals in the Nucleus Accumbens, VTA and vmPFC underpin extinction learning from omitted threats" seems misleading and unsuitable. First, this manuscript does not measure dopaminergic signals in these structures. In this study, L-DOPA is administered systemically before fear extinction and changes in BOLD signals are examined in the Nucleus Accumbens, VTA and vmPFC during extinction learning. Finding a difference in the BOLD signal in these brain structures between the L-DOPA and placebo groups does not necessarily mean that these differences are due to the direct action of L-DOPA on dopaminergic signaling in these structures. The changes in the BOLD signal in these structures can be driven indirectly by the effect of L-DOPA in another brain structure. Secondly, "underpin" which almost suggests a causal role sounds too strong considering the methods used in this manuscript and also the lack of behavioral effect of the L-DOPA treatment on extinction learning. Finally, the title emphasizes extinction learning from omitted threats and hence the reader expects that the study finds omitted threat (US omission) related activity in all the structures mentioned (nucleus accumbens, VTA and vmPFC); however, the BOLD signal in the vmPFC in the L-DOPA group increases during the CS+ in the late trials of extinction when conditioned fear responding to CS+ is significantly decreased (indicating successful extinction learning); and not during the time period of US omission. Considering these points, it would be good to change the title to more accurately reflect the results of this manuscript.

o A key concern is the lack of behavioral and psychophysiological effects of L-DOPA on extinction learning and memory. At minimum, this has serious consequences for the interpretation of the results.

– In Figure 1, L-DOPA administration does not have any significant effects during extinction learning and retention test in any of the measures examined.

– In addition, when US expectancy ratings were fitted with a Rescorla-Wager-Pearce-Hall-Hybrid model, the fitted prediction error (as a measure of expectancy violation), associability (as a measure of prediction error-guided surprise) and extinction learning rate also did not differ between the two groups indicating a lack of significant effect of L-DOPA on any of the measures.

– If there is no significant effect during extinction learning and also during the extinction retention test measuring extinction memory, I fail to see how the authors relate the changes observed in BOLD activity to extinction learning and memory. The only significant behavioral effect observed in the L-DOPA group is the decrease in the differential SCRs following reinstatement (see Figure 2). This finding serves to argue that there is a behavioral effect of L-DOPA treatment on extinction learning. However, looking at Figure S1, I do not agree with this interpretation of the reinstatement results. Although L-DOPA group shows slightly lower SCRs to CS+ following reinstatement, there seems to have no significant difference between the L-DOPA and placebo groups in their responding to CS+. One would expect to observe a significant difference in SCRs to CS+ if there is a significant effect on the strength of extinction memory. Rather, the decrease in the difference between the SCRs to CS+ and CS- in L-DOPA group seems largely due to an increase in the SCRs to CS-, as well. Therefore, it seems like the L-DOPA treatment prior to extinction learning actually resulted in decreased discrimination between the CS+ and CS- following reinstatement. This finding suggests that, contrary to expectations, the L-DOPA treatment somehow resulted in fear generalization (hence decreased safety learning) after reinstatement when measured with threat and safety discrimination. In addition, the neural mechanisms underlying differences in the fear responding of the two groups following reinstatement are also not clear. Decreased conditioned fear responses after reinstatement might also reflect an effect of L-DOPA on the strength of fear memory rather than extinction memory. To state that L-DOPA treatment enhances extinction memory, we should see a significant decrease in SCRs to CS+ during the extinction retention test in the L-DOPA group, which we fail to see in this manuscript.

• The reviewers highlighted some major concerns regarding the framing and experimental design

o There seems to be a major problem with the design of the behavioral part of the experiments. The subjects are fear conditioned in context A on day 1 and receive an extinction learning session in context B on day 2. On day 3, they receive a retention test to measure extinction memory however this test is performed not in the extinction context (context B) but rather in a new context (mix of contexts A and B). I fail to see the rationale of using a new context (particularly having parts of context A) during the retention test if the aim is to measure retention of extinction memory. It is well established that extinction is a context dependent learning. If subjects receive extinction learning in context B, they should be tested again in context B to test for retention of extinction memory. By changing the context on day 3 (and particularly by adding parts of context A), the authors are actually testing fear renewal but not retention of extinction memory. This might partly explain why they do not find a significant effect of L-DOPA treatment on extinction memory during retention test compared to previous studies that found significant enhancement of consolidation and retention of extinction memories following L-DOPA administration.

o Since the test on day 3 is measuring fear renewal rather than retrieval of extinction memory, it is not clear to me what the correlation between the vmPFC activity during extinction learning on day 2 and conditioned fear responses on day 3 reflects in Figure 3b.

• Key methodological details are lacking.

• The authors need to implement a consistent strategy for mitigating alpha inflation (i.e. correcting for multiple comparisons).

• The Discussion needs a sober and frank evaluation of the study limitations and the strength of the observed statistical effects, many of which were quite modest.

• The manuscript needs to be much more carefully copyedited.

[Editors' note: further revisions were suggested prior to acceptance, as described below.]

Thank you for resubmitting your work entitled "L-DOPA modulates activity in the vmPFC, Nucleus Accumbens and VTA during threat extinction learning in humans" for further consideration by *eLife*. Your revised article has been reviewed by 2 peer reviewers and the evaluation has been overseen by Dr. Frank as the Senior Editor, and Dr. Shackman as the Reviewing Editor.

The manuscript has been improved but there are some remaining issues that need to be addressed, as outlined below:

Reviewer #1 underscores the importance of further tempering the claims, and providing a more comprehensive discussion of alternative explanations for the results.

Reviewer #2 highlights one additional analysis to be performed.

Both reviewers make some small but important additional suggestions.

*Reviewer #1:*

The authors addressed the points previously raised be the reviewers. The major concerns still remain despite elaborate arguments by the authors that are, in part, supported with some additional statistical arguments. Specifically, they try to circumvent the lack of behavioural and psychophysiological effects during extinction training and retrieval by focusing on a decrease in the differential SCRs following reinstatement. As outlined previously by the reviewers, this effect, however, is also not very convincing because this effect seems to be due to a small increase in the SCRs to the CS- and a small decrease in the SCR to the CS+ (see Figure S1) pointing towards reduced CS discrimination (i.e. enhanced fear generalisation) which is opposite of what is reported so far for L-DOPA induced faciliation of fear exitnction(e.g. Haaker et al., 2013, Gerlicher et al., 2019). Therefore, in the present study the observed activity changes in key brain areas of the extinction neurocircuitry during extinction training do not seem to relate to facilitation of fear extinction (see all previous comments). Along these lines, the final conclusion that the authors provide a mechanistic perspective to augment extinction learning by dopaminergic enhancement in humans is not supported by the data. Please deamphasize. However, what the authors show is that neuronal activation in key brain areas of the extinction neurocircuitry is modulated by systemic L-DOPA which has been previously shown to facilitate fear extinction in healthy subjects. Please revise the abstract, results, discussion and final conclusions accordingly.

*Reviewer #2:*

In their revised manuscript, the authors provide careful responses to the majority of the points that I addressed in my previous review, and also to the compiled objections of the reviewer group:

The authors now mainly restrict the arguments to the role of L-DOPA administration instead of a "dopaminergic transmission".

Also some of the previously reported results are restricted now. This is partly due to data analyses accounting for multiple testing. Results are described more precisely now E.g., the restriction of behavioral (SCR) effects to the difference after reinstatement is stressed.

---

## [Author Response]

Essential revisions:• The reviewers highlighted some major concerns regarding the framing and interpretation of the results (as described below, and in the specific comments on the Results/Discussion, below)o Title – The title "Dopaminergic signals in the Nucleus Accumbens, VTA and vmPFC underpin extinction learning from omitted threats" seems misleading and unsuitable. First, this manuscript does not measure dopaminergic signals in these structures. In this study, L-DOPA is administered systemically before fear extinction and changes in BOLD signals are examined in the Nucleus Accumbens, VTA and vmPFC during extinction learning. Finding a difference in the BOLD signal in these brain structures between the L-DOPA and placebo groups does not necessarily mean that these differences are due to the direct action of L-DOPA on dopaminergic signaling in these structures. The changes in the BOLD signal in these structures can be driven indirectly by the effect of L-DOPA in another brain structure. Secondly, "underpin" which almost suggests a causal role sounds too strong considering the methods used in this manuscript and also the lack of behavioral effect of the L-DOPA treatment on extinction learning. Finally, the title emphasizes extinction learning from omitted threats and hence the reader expects that the study finds omitted threat (US omission) related activity in all the structures mentioned (nucleus accumbens, VTA and vmPFC); however, the BOLD signal in the vmPFC in the L-DOPA group increases during the CS+ in the late trials of extinction when conditioned fear responding to CS+ is significantly decreased (indicating successful extinction learning); and not during the time period of US omission. Considering these points, it would be good to change the title to more accurately reflect the results of this manuscript.

We understand the reviewers’ suggestions to change the title, even though “dopaminergic” describes dopamine-related signals (and not dopamine release itself). We changed the title accordingly to: “L-DOPA modulates activity in the vmPFC, Nucleus Accumbens and VTA during threat extinction learning in humans.”

o A key concern is the lack of behavioral and psychophysiological effects of L-DOPA on extinction learning and memory. At minimum, this has serious consequences for the interpretation of the results.

Our data suggests that the administration of L-DOPA enhances vmPFC signals, which mediate lower conditioned threat responses (measured as differential SCRs) during retrieval test (tested in a mediation analysis). This result is in line with a previously reported mediation effect of vmPFC activity (after extinction learning) by L-DOPA onto lower conditioned threat responses (measured as differential SCRs) during retrieval test (Gerlicher et al. 2018 Nature Communications). We, however, understand the reviewers’ comment that administration of L-DOPA in our study does not result in an (expected) univariate decrease, but a multivariate (see mediation analysis) effect of L-DOPA on behavioral and psychophysiological threat responses. Hence, we have phrased the interpretation of our results more carefully and explicitly state the absence of univariate effects in the retrieval test (for example page 20ff, discussion):

“The behavioural effects of pre-extinction administration of L-DOPA in our study were weaker when compared to studies that employed post-extinction administration [25–27, 41]. […] Future studies might employ more fine-grained explicit ratings of threat expectancy or use other psychophysiological such as pupilsize or startle responses[52, 53].”

– In Figure 1, L-DOPA administration does not have any significant effects during extinction learning and retention test in any of the measures examined.

Yes, we agree with the reviewer that the univariate effect of L-DOPA during extinction learning and retention-test is absent, which is illustrated in Figure1 and which we state within the manuscript (for example Results section page 7, description of extinction learning: “We found no statistical evidence that would support a difference between groups in US expectancy ratings or SCR”. As we outlined above, it might well be that the effect of post extinction administration of LDOPA (i.e. targeting consolidation) exerts stronger effects. Furthermore, we also found that L-DOPA enhances individual extinction learning success (see mediation analysis), which means that univariate effects of L-DOPA might not characterize the mechanisms by which L-DOPA works. (as we state on page 18: “L-DOPA might have the potential to improve (otherwise low, see parameter estimates in the Placebo group, figure 3 B)) vmPFC activity during extinction learning. Hence, rather than enhancing extinction learning per se, L-DOPA administration seems to augment vmPFC responses that accompany decreased US expectancy.”)

– In addition, when US expectancy ratings were fitted with a Rescorla-Wager-Pearce-Hall-Hybrid model, the fitted prediction error (as a measure of expectancy violation), associability (as a measure of prediction error-guided surprise) and extinction learning rate also did not differ between the two groups indicating a lack of significant effect of L-DOPA on any of the measures.

Yes, we agree with the reviewers that we found no support of a difference between L-DOPA and Placebo in modelled terms of expectancy violation (Results section page 7:)

“The fitted prediction error […], associability […] and learning rate did not differ between groups_._”

– If there is no significant effect during extinction learning and also during the extinction retention test measuring extinction memory, I fail to see how the authors relate the changes observed in BOLD activity to extinction learning and memory.

While a univariate difference in behavioral outcomes is missing, we provide a neuropharmacological mediation effect of L-DOPA (on vmPFC responses) on psychophysiological outcomes in the retrieval test. Our data suggest that L-DOPA enhances BOLD responses in the vmPFC, which mediates diminished retrieval of conditioned threat. This mediation effect is based on a previously reported effect of L-DOPA that is administered after extinction learning (Gerlicher et al. 2018 Nature Communications). Hence, we present in fact an association between L-DOPA administration and differences in BOLD signal between groups, which is linked to psychophysiological changes during retrieval test.

The only significant behavioral effect observed in the L-DOPA group is the decrease in the differential SCRs following reinstatement (see Figure 2). This finding serves to argue that there is a behavioral effect of L-DOPA treatment on extinction learning. However, looking at Figure S1, I do not agree with this interpretation of the reinstatement results. Although L-DOPA group shows slightly lower SCRs to CS+ following reinstatement, there seems to have no significant difference between the L-DOPA and placebo groups in their responding to CS+. One would expect to observe a significant difference in SCRs to CS+ if there is a significant effect on the strength of extinction memory. Rather, the decrease in the difference between the SCRs to CS+ and CS- in L-DOPA group seems largely due to an increase in the SCRs to CS-, as well. Therefore, it seems like the L-DOPA treatment prior to extinction learning actually resulted in decreased discrimination between the CS+ and CS- following reinstatement. This finding suggests that, contrary to expectations, the L-DOPA treatment somehow resulted in fear generalization (hence decreased safety learning) after reinstatement when measured with threat and safety discrimination. In addition, the neural mechanisms underlying differences in the fear responding of the two groups following reinstatement are also not clear. Decreased conditioned fear responses after reinstatement might also reflect an effect of L-DOPA on the strength of fear memory rather than extinction memory. To state that L-DOPA treatment enhances extinction memory, we should see a significant decrease in SCRs to CS+ during the extinction retention test in the L-DOPA group, which we fail to see in this manuscript.

We do not fully agree with this comment.

1. The effect of L-DOPA on the conditioned responses after reinstatement is an additional effect to the aforementioned mediation by vmPFC responses. Hence, we argue that the effect of L-DOPA is based on the mediation effect in addition to the difference after reinstatement.

2. Based on our statistical comparisons of responses after reinstatement, we state in the manuscript that we found lower *differential* CS responses (i.e., difference between CS+ and CS-) after reinstatement (page 9: “…indicating lower CS discrimination in the L-DOPA group when compared to the Placebo controls after the reinstatement procedure.”) The reviewer is right that we found nominally lower responses to the CS+ and nominally higher responses to the CS- in the L-DOPA group when compared to the Placebo group. The nominal difference between both groups are larger for the decrease in CS+ (difference 0.048) when compared to the increase in CS- responses (the difference is 0.044). But importantly these nominal differences between groups in response to the CS+ or the CS- are not supported by two-sided unpaired t-tests (CS+ t=1.4, p=0.185; CS- t=1.2, p=0.205). Hence, the idea that L-DOPA might foster generalization of responses or decreased safety learning is not supported by our data.

We however understand the comment of the reviewers that we should include additional data and analyses that underline that the effect after reinstatement is based on lower differential responses in the L-DOPA group (when compared to Placebo). We now explicitly state in the revised manuscript on page 9: “Post-hoc comparisons of CS+ and CS- responses between groups did not support a difference between L-DOPA and Placebo (p>0.19, see Table S11). […] our analyses suggest that L-DOPA administration during extinction training reduced differential threat responses after reinstatement.”

In addition, we have added Table S11 with two-sided unpaired comparisons of CS+ and CS- responses after reinstatement between groups.

• The reviewers highlighted some major concerns regarding the framing and experimental design.o There seems to be a major problem with the design of the behavioral part of the experiments. The subjects are fear conditioned in context A on day 1 and receive an extinction learning session in context B on day 2. On day 3, they receive a retention test to measure extinction memory however this test is performed not in the extinction context (context B) but rather in a new context (mix of contexts A and B). I fail to see the rationale of using a new context (particularly having parts of context A) during the retention test if the aim is to measure retention of extinction memory. It is well established that extinction is a context dependent learning. If subjects receive extinction learning in context B, they should be tested again in context B to test for retention of extinction memory. By changing the context on day 3 (and particularly by adding parts of context A), the authors are actually testing fear renewal but not retention of extinction memory. This might partly explain why they do not find a significant effect of L-DOPA treatment on extinction memory during retention test compared to previous studies that found significant enhancement of consolidation and retention of extinction memories following L-DOPA administration.

The reviewers are right that renewal phenomena contribute to the responses during the retrieval test on day3. This design (including renewal) builds upon previously reported effects of L-DOPA. L-DOPA (administered after extinction learning) has been found to decrease responses when participants were tested in both, the acquisition context (i.e., ABA renewal, Haaker et al. 2013 PNAS), as well as within the extinction context (ABB design, Gerlicher et al. 2018 Nature Communications). Hence, the employment of a mixture of context A and B during the retrieval test (which is based on a previous publication by Andreatta et al. Behavior Therapy 2015) allows to test if L-DOPA reduces threat responses in an ambiguous generalization context. The reviewers are right that we thereby probe process that underlie renewal, namely contextual generalization (see Vervliet et al. 2013 Biol Psychol). And we apologize that we have not stated this explicitly. We state this in the revised Results section (page 8): “Retrieval was tested on day 3 within an generalization context that consisted of a mixture of the acquisition (context A) and extinction context (context B) [34], which also involves contextual renewal of conditioned threat responses [35]” and the methods description on page 22: “Retrieval test (including reinstatement procedure) was conducted in a 50/50-mixture of context A and B in order to examine contextual generalization [34], which also involves contextual renewal of conditioned threat responses[35].”.

Theories and experimental studies on renewal (and in particular contextual generalization) provide evidence that renewal is not a differential (i.e. “all or nothing”) situation in which either fear or extinction memories are retrieved. Rather, retrieval of both, acquisition and extinction memory play a role during renewal (see Vervliet et al. 2013 Biol Psychol, Bouton Biol Psychiatry 2002). In line with this theoretical account is the finding that the vmPFC plays a role in renewal situations, even when the behavioural output of conditioned responses is stronger than their inhibition. For example, renewal (ABA) and extinction retrieval (ABB) both involve projections from the ventral hippocampus to the infralimbic cortex (rodents vmPFC homologue, Wang, Jin and Maren Neuobiol Learn Mem 2016). Pathway specific activation of projections from the hippocampus to the infralimbic cortex (rodents vmPFC homologue) diminishes behavioral renewal outside the extinction context (Vasquez Neuobiol Learn Mem 2019). Furthermore, ABC renewal (which might be probed in our design) is not as strong as ABA renewal (e.g., Harris et al. 2000), which further indicates that inhibition of conditioned responses plays a role in contextual renewal. We however see the reviewers point that the retrieval test probes threat, as well as extinction memories. In order to make this clear, we have explicitly termed the test phase on day 3 “retrieval test” (instead of “extinction retrieval test”). In order to follow the reviewers’ suggestion, we have further replaced “extinction retrieval” with “retrieval” at several parts in the manuscript, for example page 11 (results): “[…] we tested if this difference in the right vmPFC activity is related to individual differences in the retrieval of conditioned threat responses.” and “[…] enhanced vmPFC activity is associated with reduced retrieval of differential threat responses (measured as SCR) 24 hours later”.

o Since the test on day 3 is measuring fear renewal rather than retrieval of extinction memory, it is not clear to me what the correlation between the vmPFC activity during extinction learning on day 2 and conditioned fear responses on day 3 reflects in Figure 3b.

The mediation analysis in our results is actually based on a study that employed extinction memory retrieval (i.e. ABB design, Gerlicher et al. Nature Communications 2018). The authors found an effect of L-DOPA on extinction memory retrieval that is mediated by vmPFC activity. As the reviewers pointed out, the design initially used by Gerlicher allows to unambitiously examine extinction memory retrieval. We found the same association between responses within the retrieval test in our design. Hence, the same mediation effect of LDOPA onto conditioned responses by the vmPFC in both studies further suggests that L-DOPA affects extinction memory retrieval in our design.

• Key methodological details are lacking.• The authors need to implement a consistent strategy for mitigating alpha inflation (i.e. correcting for multiple comparisons).

We understand this and we have added to the already applied strategy to mitigate alpha inflation the correction of exploratory control analyses. For example, we could thereby clarify the results on fear ratings on page 7: “Exploratory analyses suggested that this effect might be driven by lower ratings to the CS+ and the extinction context (presented as the ITI) in the L-DOPA group, but none of these comparisons survived correction for multiple testing (p-values (FWE)> 0.256”).

• The Discussion needs a sober and frank evaluation of the study limitations and the strength of the observed statistical effects, many of which were quite modest.

We have adjusted the interpretation of our results (Results section and discussion) to address this comment and rephrased the limitations of our study (within a new paragraph in the discussion).

“Limitations of this study

This pharmacological fMRI study in human volunteers is only suited to draw inferences on blood-oxygen-level-dependent signals as a function of L-DOPA administration. […] Future studies that further investigate L-DOPA as a novel augmentation strategy for the therapy of anxiety related disorders (in which extinction mechanisms are only one part of the process) are warranted.”

• The manuscript needs to be much more carefully copyedited.

All authors have carefully copyedited the manuscript together with two scientists that are very fluent in English.

[Editors' note: further revisions were suggested prior to acceptance, as described below.]

Reviewer #1:The authors addressed the points previously raised be the reviewers. The major concerns still remain despite elaborate arguments by the authors that are, in part, supported with some additional statistical arguments. Specifically, they try to circumvent the lack of behavioural and psychophysiological effects during extinction training and retrieval by focusing on a decrease in the differential SCRs following reinstatement. As outlined previously by the reviewers, this effect, however, is also not very convincing because this effect seems to be due to a small increase in the SCRs to the CS- and a small decrease in the SCR to the CS+ (see Figure S1) pointing towards reduced CS discrimination (i.e. enhanced fear generalisation) which is opposite of what is reported so far for L-DOPA induced faciliation of fear exitnction(e.g. Haaker et al., 2013, Gerlicher et al., 2019). Therefore, in the present study the observed activity changes in key brain areas of the extinction neurocircuitry during extinction training do not seem to relate to facilitation of fear extinction (see all previous comments). Along these lines, the final conclusion that the authors provide a mechanistic perspective to augment extinction learning by dopaminergic enhancement in humans is not supported by the data. Please deamphasize. However, what the authors show is that neuronal activation in key brain areas of the extinction neurocircuitry is modulated by systemic L-DOPA which has been previously shown to facilitate fear extinction in healthy subjects. Please revise the abstract, results, discussion and final conclusions accordingly.

We thank the reviewer for the evaluation of our work. We have mentioned in the previous revision that our mediation analysis actually provides evidence on how “activity changes in key brain areas of the extinction neurocircuitry during extinction training […] relate to facilitation of fear extinction”, to quote what the reviewer misses in our manuscript. Namely, we found that the activity in the vmPFC mediates the effect of LDOPA on reduced SCR in the retrieval test, which is similar to the effect as reported by Gerlicher et al. 2019. This effect is highlighted in the manuscript (and not merely the effect found after reinstatement).

Additionally, our finding of reduced differential CS response in the L-DOPA group relative to the Placebo group after reinstatement is not contrary to what has been found in Gerlicher et al. 2019 or Haaker et al. 2013. For example, in Haaker et al. 2013, we found that “The placebo group showed significantly larger differential SCRs (CS+ > CS−) in context A (test 2) than in context B (test 1) on day 2 […] This pronounced renewal effect was significantly attenuated in the L-dopa group (cue by context by group interaction: F1,31 = 5.23, P = 0.029). Further inspection suggested that the interaction was mainly due to relatively decreased responding to the CS+ in context A (post hoc t test: t1,31 = 1.81, P = 0.082”. Hence, there is a differential effect on the SCR between groups, similar to what we find after reinstatement (Gerlicher et al. also report a stimulus * group interaction)). In our current study, we report a similar stimulus*group interaction and if we would compare CS+ responses between groups, there would be a trend for lower responses in the L-DOPA group (one sided t-test t(40)=1,348 p=0.093). However, since the ANOVA revealed a CS*group interaction it is more appropriate to compare differential CS responses.

Nevertheless, we deemphasized our results and now state in the abstract (page 1):

“Dopaminergic enhancement via administration of L-DOPA (vs. Placebo) was associated with reduced retention of differential psychophysiological threat responses at later test, which was mediated by activity in the ventromedial prefrontal cortex that was specific to extinction learning.”

And in the result section (page 9): “While our analysis revealed a difference between groups in differential SCRs, there was no strong support for difference between groups in CS+ or CS- responses.”

And in the discussion (page 17):

“Additionally, we found reduced differential SCRs (CS+> CS-, without strong support for a reduction in CS+ responses, only) in the L-DOPA group when compared to Placebo after the reinstatement one day after extinction learning.”

Reviewer #2:In their revised manuscript, the authors provide careful responses to the majority of the points that I addressed in my previous review, and also to the compiled objections of the reviewer group:The authors now mainly restrict the arguments to the role of L-DOPA administration instead of a "dopaminergic transmission".Also some of the previously reported results are restricted now. This is partly due to data analyses accounting for multiple testing. Results are described more precisely now E.g., the restriction of behavioral (SCR) effects to the difference after reinstatement is stressed.

We are thankful for the suggestion of the reviewer and the suggested changes helped to clarify our main message in the manuscript.